# Annealing Between Distributions by Averaging Moments

**Roger Grosse**
Comp. Sci. & AI Lab
MIT
Cambridge, MA 02139

**Chris J. Maddison**
Dept. of Computer Science
University of Toronto
Toronto, ON M5S 3G4

**Ruslan Salakhutdinov**
Depts. of Statistics and Comp. Sci.,
University of Toronto
Toronto, ON M5S 3G4, Canada

## Abstract

Many powerful Monte Carlo techniques for estimating partition functions, such as annealed importance sampling (AIS), are based on sampling from a sequence of intermediate distributions which interpolate between a tractable initial distribution and the intractable target distribution. The near-universal practice is to use geometric averages of the initial and target distributions, but alternative paths can perform substantially better. We present a novel sequence of intermediate distributions for exponential families defined by averaging the moments of the initial and target distributions. We analyze the asymptotic performance of both the geometric and moment averages paths and derive an asymptotically optimal piecewise linear schedule. AIS with moment averaging performs well empirically at estimating partition functions of restricted Boltzmann machines (RBMs), which form the building blocks of many deep learning models.

## 1   Introduction

Many generative models are defined in terms of an unnormalized probability distribution, and computing the probability of a state requires computing the (usually intractable) partition function. This is problematic for model selection, since one often wishes to compute the probability assigned to held-out test data. While partition function estimation is intractable in general, there has been extensive research on variational [1, 2, 3] and sampling-based [4, 5, 6] approximations. In the context of model comparison, annealed importance sampling (AIS) [4] is especially widely used because given enough computing time, it can provide high-accuracy estimates. AIS has enabled precise quantitative comparisons of powerful generative models in image statistics [7, 8] and deep learning [9, 10, 11]. Unfortunately, applying AIS in practice can be computationally expensive and require laborious hand-tuning of annealing schedules. Because of this, many generative models still have not been quantitatively compared in terms of held-out likelihood.

AIS requires defining a sequence of intermediate distributions which interpolate between a tractable initial distribution and the intractable target distribution. Typically, one uses geometric averages of the initial and target distributions. Tantalizingly, [12] derived the optimal paths for some toy models in the context of path sampling and showed that they vastly outperformed geometric averages. However, as choosing an optimal path is generally intractable, geometric averages still predominate.

In this paper, we present a theoretical framework for evaluating alternative paths. We propose a novel path defined by averaging moments of the initial and target distributions. We show that geometric averages and moment averages optimize different variational objectives, derive an asymptotically optimal piecewise linear schedule, and analyze the asymptotic performance of both paths. Our proposed path often outperforms geometric averages at estimating partition functions of restricted Boltzmann machines (RBMs).

## 2 Estimating Partition Functions

Suppose we have a probability distribution $p_b(\mathbf{x}) = f_b(\mathbf{x})/\mathcal{Z}_b$ defined on a space $\mathcal{X}$, where $f_b(\mathbf{x})$ can be computed efficiently for a given $\mathbf{x} \in \mathcal{X}$, and we are interested in estimating the partition function $\mathcal{Z}_b$. Annealed importance sampling (AIS) is an algorithm which estimates $\mathcal{Z}_b$ by gradually changing, or "annealing," a distribution. In particular, one must specify a sequence of $K+1$ intermediate distributions $p_k(\mathbf{x}) = f_k(\mathbf{x})/\mathcal{Z}_k$ for $k = 0, \ldots K$, where $p_a(\mathbf{x}) = p_0(\mathbf{x})$ is a tractable initial distribution, and $p_b(\mathbf{x}) = p_K(\mathbf{x})$ is the intractable target distribution. For simplicity, assume all distributions are strictly positive on $\mathcal{X}$. For each $p_k$, one must also specify an MCMC transition operator $T_k$ (e.g. Gibbs sampling) which leaves $p_k$ invariant. AIS alternates between MCMC transitions and importance sampling updates, as shown in Alg 1.

The output of AIS is an unbiased estimate $\hat{\mathcal{Z}}_b$ of $\mathcal{Z}_b$. Remarkably, unbiasedness holds even in the context of *non-equilibrium* samples along the chain [4, 13]. However, unless the intermediate distributions and transition operators are carefully chosen, $\hat{\mathcal{Z}}_b$ may have high variance and be far from $\mathcal{Z}_b$ with high probability.

The mathematical formulation of AIS leaves much flexibility for choosing intermediate distributions. However, one typically defines a path $\gamma : [0,1] \rightarrow \mathcal{P}$ through some family $\mathcal{P}$ of distributions. The intermediate distributions

---

**Algorithm 1** Annealed Importance Sampling

> **for** $i = 1$ to $M$ **do**
>     $\mathbf{x}_0 \leftarrow$ sample from $p_0(\mathbf{x})$
>     $w^{(i)} \leftarrow \mathcal{Z}_a$
>     **for** $k = 1$ to $K$ **do**
>         $w^{(i)} \leftarrow w^{(i)} \frac{f_k(\mathbf{x}_{k-1})}{f_{k-1}(\mathbf{x}_{k-1})}$
>         $\mathbf{x}_k \leftarrow$ sample from $T_k\left(\mathbf{x} \,|\, \mathbf{x}_{k-1}\right)$
>     **end for**
> **end for**
> **return** $\hat{\mathcal{Z}}_b = \sum_{i=1}^M w^{(i)}/M$

---

$p_k$ are chosen to be points along this path corresponding to a *schedule* $0 = \beta_0 < \beta_1 < \ldots < \beta_K = 1$. One typically uses the geometric path $\gamma_{GA}$, defined in terms of geometric averages of $p_a$ and $p_b$:

$$p_\beta(\mathbf{x}) = f_\beta(\mathbf{x})/\mathcal{Z}(\beta) = f_a(\mathbf{x})^{1-\beta} f_b(\mathbf{x})^\beta / \mathcal{Z}(\beta). \tag{1}$$

Commonly, $f_a$ is the uniform distribution, and (1) reduces to $p_\beta(\mathbf{x}) = f_b(\mathbf{x})^\beta / \mathcal{Z}(\beta)$. This motivates the term "annealing," and $\beta$ resembles an inverse temperature parameter. As in simulated annealing, the "hotter" distributions often allow faster mixing between modes which are isolated in $p_b$.

AIS is closely related to a broader family of techniques for posterior inference and partition function estimation, all based on the following identity from statistical physics:

$$\log \mathcal{Z}_b - \log \mathcal{Z}_a = \int_0^1 \mathbb{E}_{\mathbf{x} \sim p_\beta} \left[ \frac{\mathrm{d}}{\mathrm{d}\beta} \log f_\beta(\mathbf{x}) \right] \mathrm{d}\beta. \tag{2}$$

Thermodynamic integration [14] estimates (2) using numerical quadrature, and path sampling [12] does so with Monte Carlo integration. The weight update in AIS can be seen as a finite difference approximation. Tempered transitions [15] is a Metropolis-Hastings proposal operator which heats up and cools down the distribution, and computes an acceptance ratio by approximating (2).

The choices of a path and a schedule are central to all of these methods. Most work on adapting paths has focused on tuning schedules along a geometric path [15, 16, 17]. [15] showed that the geometric *schedule* was optimal for annealing the scale parameter of a Gaussian, and [16] extended this result more broadly. The aim of this paper is to propose, analyze, and evaluate a novel alternative to $\gamma_{GA}$ based on averaging moments of the initial and target distributions.

## 3 Analyzing AIS Paths

When analyzing AIS, it is common to assume *perfect transitions*, i.e. that each transition operator $T_k$ returns an independent and exact sample from the distribution $p_k$ [4]. This corresponds to the (somewhat idealized) situation where the Markov chains mix quickly. As Neal [4] pointed out, assuming perfect transitions, the Central Limit Theorem shows that the samples $w^{(i)}$ are approximately log-normally distributed. In this case, the variances $\mathrm{var}(w^{(i)})$ and $\mathrm{var}(\log w^{(i)})$ are both monotonically related to $\mathbb{E}[\log w^{(i)}]$. Therefore, our analysis focuses on $\mathbb{E}[\log w^{(i)}]$.

Assuming perfect transitions, the expected log weights are given by:

$$\mathbb{E}[\log w^{(i)}] = \log \mathcal{Z}_a + \sum_{k=0}^{K-1} \mathbb{E}_{p_k}[\log f_{k+1}(\mathbf{x}) - \log f_k(\mathbf{x})] = \log \mathcal{Z}_b - \sum_{k=0}^{K-1} \mathrm{D}_{\mathrm{KL}}(p_k \| p_{k+1}). \tag{3}$$

In other words, each $\log w^{(i)}$ can be seen as a biased estimator of $\log \mathcal{Z}_b$, where the bias $\delta = \log \mathcal{Z}_b - \mathbb{E}[\log w^{(i)}]$ is given by the sum of KL divergences $\sum_{k=0}^{K-1} \mathrm{D_{KL}}(p_k \| p_{k+1})$.

Suppose $\mathcal{P}$ is a family of probability distributions parameterized by $\boldsymbol{\theta} \in \Theta$, and the $K+1$ distributions $p_0, \ldots, p_K$ are chosen to be linearly spaced along a path $\gamma : [0, 1] \to \mathcal{P}$. Let $\boldsymbol{\theta}(\beta)$ represent the parameters of the distribution $\gamma(\beta)$. As $K$ is increased, the bias $\delta$ decays like $1/K$, and the asymptotic behavior is determined by a functional $\mathcal{F}(\gamma)$.

**Theorem 1.** *Suppose $K+1$ distributions $p_k$ are linearly spaced along a path $\gamma$. Assuming perfect transitions, if $\boldsymbol{\theta}(\beta)$ and the Fisher information matrix $\mathbf{G}_{\boldsymbol{\theta}}(\beta) = \mathrm{cov}_{\mathbf{x} \sim p_{\boldsymbol{\theta}}}(\nabla_{\boldsymbol{\theta}} \log p_{\boldsymbol{\theta}}(\mathbf{x}))$ are continuous and piecewise smooth, then as $K \to \infty$ the bias $\delta$ behaves as follows:*

$$K\delta = K \sum_{k=0}^{K-1} \mathrm{D_{KL}}(p_k \| p_{k+1}) \to \mathcal{F}(\gamma) \equiv \frac{1}{2} \int_0^1 \dot{\boldsymbol{\theta}}(\beta)^T \mathbf{G}_{\boldsymbol{\theta}}(\beta) \dot{\boldsymbol{\theta}}(\beta) d\beta, \qquad (4)$$

*where $\dot{\boldsymbol{\theta}}(\beta)$ represents the derivative of $\boldsymbol{\theta}$ with respect to $\beta$. [See supplementary material for proof.]*

This result reveals a relationship with path sampling, as [12] showed that the variance of the path sampling estimator is proportional to the same functional. One useful result from their analysis is a derivation of the optimal schedule along a given path. In particular, the value of $\mathcal{F}(\gamma)$ using the optimal schedule is given by $\ell(\gamma)^2/2$, where $\ell$ is the Riemannian path length defined by

$$\ell(\gamma) = \int_0^1 \sqrt{\dot{\boldsymbol{\theta}}(\beta)^T \mathbf{G}_{\boldsymbol{\theta}}(\beta) \dot{\boldsymbol{\theta}}(\beta)} d\beta. \qquad (5)$$

Intuitively, the optimal schedule allocates more distributions to regions where $p_\beta$ changes quickly. While [12] derived the optimal paths and schedules for some simple examples, they observed that this is intractable in most cases and recommended using geometric paths in practice.

The above analysis assumes perfect transitions, which can be unrealistic in practice because many distributions have separated modes between which mixing is difficult. As Neal [4] observed, in such cases, AIS can be viewed as having two sources of variance: that caused by variability within a mode, and that caused by misallocation of samples between modes. The former source of variance is well modeled by the perfect transitions analysis and can be made small by adding more intermediate distributions. The latter, however, can persist even with large numbers of intermediate distributions. While our theoretical analysis assumes perfect transitions, our proposed method often gave substantial improvement empirically in situations with poor mixing.

## 4 Moment Averaging

As discussed in Section 2, the typical choice of intermediate distributions for AIS is the geometric averages path $\gamma_{GA}$ given by (1). In this section, we propose an alternative path for exponential family models. An exponential family model is defined as

$$p(\mathbf{x}) = \frac{1}{\mathcal{Z}(\boldsymbol{\eta})} h(\mathbf{x}) \exp\left(\boldsymbol{\eta}^T \mathbf{g}(\mathbf{x})\right), \qquad (6)$$

where $\boldsymbol{\eta}$ are the natural parameters and $\mathbf{g}$ are the sufficient statistics. Exponential families include a wide variety of statistical models, including Markov random fields.

In exponential families, geometric averages correspond to averaging the natural parameters:

$$\boldsymbol{\eta}(\beta) = (1 - \beta)\boldsymbol{\eta}(0) + \beta\boldsymbol{\eta}(1). \qquad (7)$$

Exponential families can also be parameterized in terms of their moments $\mathbf{s} = \mathbb{E}[\mathbf{g}(\mathbf{x})]$. For any minimal exponential family (i.e. one whose sufficient statistics are linearly independent), there is a one-to-one mapping between moments and natural parameters [18, p. 64]. We propose an alternative to $\gamma_{GA}$ called the *moment averages* path, denoted $\gamma_{MA}$, and defined by averaging the moments of the initial and target distributions:

$$\mathbf{s}(\beta) = (1 - \beta)\mathbf{s}(0) + \beta\mathbf{s}(1). \qquad (8)$$

This path exists for any exponential family model, since the set of realizable moments is convex [18]. It is unique, since $\mathbf{g}$ is unique up to affine transformation.

As an illustrative example, consider a multivariate Gaussian distribution parameterized by the mean $\boldsymbol{\mu}$ and covariance $\boldsymbol{\Sigma}$. The moments are $\mathbb{E}[\mathbf{x}] = \boldsymbol{\mu}$ and $-\frac{1}{2}\mathbb{E}[\mathbf{x}\mathbf{x}^T] = -\frac{1}{2}(\boldsymbol{\Sigma} + \boldsymbol{\mu}\boldsymbol{\mu}^T)$. By plugging these into (8), we find that $\gamma_{MA}$ is given by:

$$\boldsymbol{\mu}(\beta) = (1-\beta)\boldsymbol{\mu}(0) + \beta\boldsymbol{\mu}(1) \tag{9}$$

$$\boldsymbol{\Sigma}(\beta) = (1-\beta)\boldsymbol{\Sigma}(0) + \beta\boldsymbol{\Sigma}(1) + \beta(1-\beta)(\boldsymbol{\mu}(1) - \boldsymbol{\mu}(0))(\boldsymbol{\mu}(1) - \boldsymbol{\mu}(0))^T. \tag{10}$$

In other words, the means are linearly interpolated, and the covariances are linearly interpolated and stretched in the direction connecting the two means. Intuitively, this stretching is a useful property, because it increases the overlap between successive distributions with different means. A comparison of the two paths is shown in Figure 1.

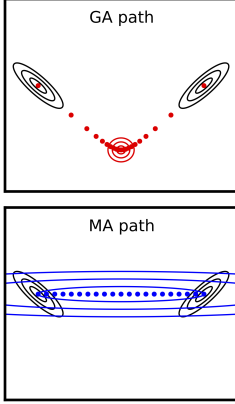

Figure 1: Comparison of $\gamma_{GA}$ and $\gamma_{MA}$ for multivariate Gaussians: intermediate distribution for $\beta = 0.5$, and $\boldsymbol{\mu}(\beta)$ for $\beta$ evenly spaced from 0 to 1.

Next consider the example of a restricted Boltzmann machine (RBM), a widely used model in deep learning. A binary RBM is a Markov random field over binary vectors $\mathbf{v}$ (the visible units) and $\mathbf{h}$ (the hidden units), and which has the distribution

$$p(\mathbf{v}, \mathbf{h}) \propto \exp\left(\mathbf{a}^T\mathbf{v} + \mathbf{b}^T\mathbf{h} + \mathbf{v}^T\mathbf{W}\mathbf{h}\right). \tag{11}$$

The parameters of the model are the visible biases $\mathbf{a}$, the hidden biases $\mathbf{b}$, and the weights $\mathbf{W}$. Since these parameters are also the natural parameters in the exponential family representation, $\gamma_{GA}$ reduces to linearly averaging the biases and the weights. The sufficient statistics of the model are the visible activations $\mathbf{v}$, the hidden activations $\mathbf{h}$, and the products $\mathbf{v}\mathbf{h}^T$. Therefore, $\gamma_{MA}$ is defined by:

$$\mathbb{E}[\mathbf{v}]_\beta = (1-\beta)\mathbb{E}[\mathbf{v}]_0 + \beta\mathbb{E}[\mathbf{v}]_1 \tag{12}$$

$$\mathbb{E}[\mathbf{h}]_\beta = (1-\beta)\mathbb{E}[\mathbf{h}]_0 + \beta\mathbb{E}[\mathbf{h}]_1 \tag{13}$$

$$\mathbb{E}[\mathbf{v}\mathbf{h}^T]_\beta = (1-\beta)\mathbb{E}[\mathbf{v}\mathbf{h}^T]_0 + \beta\mathbb{E}[\mathbf{v}\mathbf{h}^T]_1 \tag{14}$$

For many models of interest, including RBMs, it is infeasible to determine $\gamma_{MA}$ exactly, as it requires solving two often intractable problems: (1) computing the moments of $p_b$, and (2) solving for model parameters which match the averaged moments $\mathbf{s}(\beta)$. However, much work has been devoted to practical approximations [19, 20], some of which we use in our experiments with intractable models. Since it would be infeasible to moment match every $\beta_k$ even approximately, we introduce the moment averages spline (MAS) path, denoted $\gamma_{MAS}$. We choose a set of $R$ values $\beta_1, \ldots, \beta_R$ called *knots*, and solve for the natural parameters $\boldsymbol{\eta}(\beta_j)$ to match the moments $\mathbf{s}(\beta_j)$ for each knot. We then interpolate between the knots using geometric averages. The analysis of Section 4.2 shows that, under the assumption of perfect transitions, using $\gamma_{MAS}$ in place of $\gamma_{MA}$ does not affect the cost functional $\mathcal{F}$ defined in Theorem 1.

## 4.1 Variational Interpretation

By interpreting $\gamma_{GA}$ and $\gamma_{MA}$ as optimizing different variational objectives, we gain additional insight into their behavior. For geometric averages, the intermediate distribution $\gamma_{GA}(\beta)$ minimizes a weighted sum of KL divergences to the initial and target distributions:

$$p_\beta^{(GA)} = \arg\min_q (1-\beta)\mathrm{D}_{\mathrm{KL}}(q\|p_a) + \beta\mathrm{D}_{\mathrm{KL}}(q\|p_b). \tag{15}$$

On the other hand, $\gamma_{MA}$ minimizes the weighted sum of KL divergences in the reverse direction:

$$p_\beta^{(MA)} = \arg\min_q (1-\beta)\mathrm{D}_{\mathrm{KL}}(p_a\|q) + \beta\mathrm{D}_{\mathrm{KL}}(p_b\|q). \tag{16}$$

See the supplementary material for the derivations. The objective function (15) is minimized by a distribution which puts significant mass only in the "intersection" of $p_a$ and $p_b$, i.e. those regions which are likely under both distributions. By contrast, (16) encourages the distribution to be spread out in order to capture all high probability regions of both $p_a$ and $p_b$. This interpretation helps explain why the intermediate distributions in the Gaussian example of Figure 1 take the shape that they do. In our experiments, we found that $\gamma_{MA}$ often gave more accurate results than $\gamma_{GA}$ because the intermediate distributions captured regions of the target distribution which were missed by $\gamma_{GA}$.

## 4.2 Asymptotics under Perfect Transitions

In general, we found that $\gamma_{GA}$ and $\gamma_{MA}$ can look very different. Intriguingly, both paths always result in the same value of the cost functional $\mathcal{F}(\gamma)$ of Theorem 1 for any exponential family model. Furthermore, nothing is lost by using the spline approximation $\gamma_{MAS}$ in place of $\gamma_{MA}$:

**Theorem 2.** *For any exponential family model with natural parameters $\boldsymbol{\eta}$ and moments $\mathbf{s}$, all three paths share the same value of the cost functional:*

$$\mathcal{F}(\gamma_{GA}) = \mathcal{F}(\gamma_{MA}) = \mathcal{F}(\gamma_{MAS}) = \frac{1}{2}(\boldsymbol{\eta}(1) - \boldsymbol{\eta}(0))^T(\mathbf{s}(1) - \mathbf{s}(0)). \tag{17}$$

*Proof.* The two parameterizations of exponential families satisfy the relationship $\mathbf{G}_{\boldsymbol{\eta}}\dot{\boldsymbol{\eta}} = \dot{\mathbf{s}}$ [21, sec. 3.3]. Therefore, $\mathcal{F}(\gamma)$ can be rewritten as $\frac{1}{2}\int_0^1 \dot{\boldsymbol{\eta}}(\beta)^T\dot{\mathbf{s}}(\beta)\,\mathrm{d}\beta$. Because $\gamma_{GA}$ and $\gamma_{MA}$ linearly interpolate the natural parameters and moments respectively,

$$\mathcal{F}(\gamma_{GA}) = \frac{1}{2}(\boldsymbol{\eta}(1) - \boldsymbol{\eta}(0))^T \int_0^1 \dot{\mathbf{s}}(\beta)\,\mathrm{d}\beta = \frac{1}{2}(\boldsymbol{\eta}(1) - \boldsymbol{\eta}(0))^T(\mathbf{s}(1) - \mathbf{s}(0)) \tag{18}$$

$$\mathcal{F}(\gamma_{MA}) = \frac{1}{2}(\mathbf{s}(1) - \mathbf{s}(0))^T \int_0^1 \dot{\boldsymbol{\eta}}(\beta)\,\mathrm{d}\beta = \frac{1}{2}(\mathbf{s}(1) - \mathbf{s}(0))^T(\boldsymbol{\eta}(1) - \boldsymbol{\eta}(0)). \tag{19}$$

Finally, to show that $\mathcal{F}(\gamma_{MAS}) = \mathcal{F}(\gamma_{MA})$, observe that $\gamma_{MAS}$ uses the geometric path between each pair of knots $\gamma(\beta_j)$ and $\gamma(\beta_{j+1})$, while $\gamma_{MA}$ uses the moments path. The above analysis shows the costs must be equal for each segment, and therefore equal for the entire path. $\qquad\square$

This analysis shows that all three paths result in the same expected log weights asymptotically, assuming perfect transitions. There are several caveats, however. First, we have noticed experimentally that $\gamma_{MA}$ often yields substantially more accurate estimates of $\mathcal{Z}_b$ than $\gamma_{GA}$ even when the average log weights are comparable. Second, the two paths can have very different mixing properties, which can strongly affect the results. Third, Theorem 2 assumes *linear* schedules, and in principle there is room for improvement if one is allowed to tune the schedule.

For instance, consider annealing between two Gaussians $p_a = \mathcal{N}(\mu_a, \sigma)$ and $p_b = \mathcal{N}(\mu_b, \sigma)$. The optimal schedule for $\gamma_{GA}$ is a linear schedule with cost $\mathcal{F}(\gamma_{GA}) = O(d^2)$, where $d = |\mu_b - \mu_a|/\sigma$. Using a linear schedule, the moment path also has cost $O(d^2)$, consistent with Theorem 2. However, most of the cost of the path results from instability near the endpoints, where the variance changes suddenly. Using an optimal schedule, which allocates more distributions near the endpoints, the cost functional falls to $O((\log d)^2)$, which is within a constant factor of the optimal path derived by [12]. (See the supplementary material for the derivations.) In other words, while $\mathcal{F}(\gamma_{GA}) = \mathcal{F}(\gamma_{MA})$, they achieve this value for different reasons: $\gamma_{GA}$ follows an optimal schedule along a bad path, while $\gamma_{MA}$ follows a bad schedule along a near-optimal path. We speculate that, combined with the procedure of Section 4.3 for choosing a schedule, moment averages may result in large reductions in the cost functional for some models.

## 4.3 Optimal Binned Schedules

In general, it is hard to choose a good schedule for a given path. However, consider the set of *binned schedules*, where the path is divided into segments, some number $K_j$ of intermediate distributions are allocated to each segment, and the distributions are spaced linearly within each segment. Under the assumption of perfect transitions, there is a simple formula for an asymptotically optimal binned schedule which requires only the parameters obtained through moment averaging:

**Theorem 3.** *Let $\gamma$ be any path for an exponential family model defined by a set of knots $\beta_j$, each with natural parameters $\boldsymbol{\eta}_j$ and moments $\mathbf{s}_j$, connected by segments of either $\gamma_{GA}$ or $\gamma_{MA}$ paths. Then, under the assumption of perfect transitions, an asymptotically optimal allocation of intermediate distributions to segments is given by:*

$$K_j \propto \sqrt{(\boldsymbol{\eta}_{j+1} - \boldsymbol{\eta}_j)^T(\mathbf{s}_{j+1} - \mathbf{s}_j)}. \tag{20}$$

*Proof.* By Theorem 2, the cost functional for segment $j$ is $F_j = \frac{1}{2}(\boldsymbol{\eta}_{j+1} - \boldsymbol{\eta}_j)^T(\mathbf{s}_{j+1} - \mathbf{s}_j)$. Hence, with $K_j$ distributions allocated to it, it contributes $F_j/K_j$ to the total cost. The values of $K_j$ which minimize $\sum_j F_j/K_j$ subject to $\sum_j K_j = K$ and $K_j \geq 0$ are given by $K_j \propto \sqrt{F_j}$. $\qquad\square$

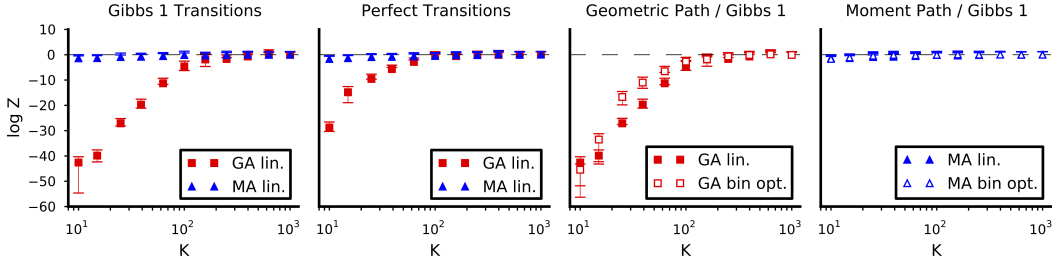

Figure 2: Estimates of $\log \mathcal{Z}_b$ for a normalized Gaussian as $K$, the number of intermediate distributions, is varied. True value: $\log \mathcal{Z}_b = 0$. Error bars show bootstrap 95% confidence intervals. (Best viewed in color.)

## 5 Experimental Results

In order to compare our proposed path with geometric averages, we ran AIS using each path to estimate partition functions of several probability distributions. For all of our experiments, we report two sets of results. First, we show the estimates of $\log \mathcal{Z}$ as a function of $K$, the number of intermediate distributions, in order to visualize the amount of computation necessary to obtain reasonable accuracy. Second, as recommended by [4], we report the effective sample size (ESS) of the weights for a large $K$. This statistic roughly measures how many independent samples one obtains using AIS.[1] All results are based on 5,000 independent AIS runs, so the maximum possible ESS is 5,000.

### 5.1 Annealing Between Two Distant Gaussians

In our first experiment, the initial and target distributions were the two Gaussians shown in Fig. 1, whose parameters are $\mathcal{N}\left(\left(\begin{smallmatrix}-10\\0\end{smallmatrix}\right), \left(\begin{smallmatrix}1 & -0.85\\-0.85 & 1\end{smallmatrix}\right)\right)$ and $\mathcal{N}\left(\left(\begin{smallmatrix}10\\0\end{smallmatrix}\right), \left(\begin{smallmatrix}1 & 0.85\\0.85 & 1\end{smallmatrix}\right)\right)$. As both distributions are normalized, $\mathcal{Z}_a = \mathcal{Z}_b = 1$. We compared $\gamma_{GA}$ and $\gamma_{MA}$ both under perfect transitions, and using the Gibbs transition operator. We also compared linear schedules with the optimal binned schedules of Section 4.3, using 10 segments evenly spaced from 0 to 1.

Figure 2 shows the estimates of $\log \mathcal{Z}_b$ for $K$ ranging from 10 to 1,000. Observe that with 1,000 intermediate distributions, all paths yielded accurate estimates of $\log \mathcal{Z}_b$. However, $\gamma_{MA}$ needed fewer intermediate distributions to achieve accurate estimates. For example, with $K = 25$, $\gamma_{MA}$ resulted in an estimate within one nat of $\log \mathcal{Z}_b$, while the estimate based on $\gamma_{GA}$ was off by 27 nats.

This result may seem surprising in light of Theorem 2, which implies that $\mathcal{F}(\gamma_{GA}) = \mathcal{F}(\gamma_{MA})$ for linear schedules. In fact, the average log weights for $\gamma_{GA}$ and $\gamma_{MA}$ were similar for all values of $K$, as the theorem would suggest; e.g., with $K = 25$, the average was -27.15 for $\gamma_{MA}$ and -28.04 for $\gamma_{GA}$. However, because the $\gamma_{MA}$ intermediate distributions were broader, enough samples landed in high probability regions to yield reasonable estimates of $\log \mathcal{Z}_b$.

### 5.2 Partition Function Estimation for RBMs

Our next set of experiments focused on restricted Boltzmann machines (RBMs), a building block of many deep learning models (see Section 4). We considered RBMs trained with three different methods: contrastive divergence (CD) [19] with one step (CD1), CD with 25 steps (CD25), and persistent contrastive divergence (PCD) [20]. All of the RBMs were trained on the MNIST handwritten digits dataset [22], which has long served as a benchmark for deep learning algorithms. We experimented both with small, tractable RBMs and with full-size, intractable RBMs.

Since it is hard to compute $\gamma_{MA}$ exactly for RBMs, we used the moments spline path $\gamma_{MAS}$ of Section 4 with the 9 knot locations $0.1, 0.2, \ldots, 0.9$. We considered the two initial distributions discussed by [9]: (1) the uniform distribution, equivalent to an RBM where all the weights and biases are set to 0, and (2) the *base rate RBM*, where the weights and hidden biases are set to 0, and the visible biases are set to match the average pixel values over the MNIST training set.

**Small, Tractable RBMs:** To better understand the behavior of $\gamma_{GA}$ and $\gamma_{MAS}$, we first evaluated the paths on RBMs with only 20 hidden units. In this setting, it is feasible to exactly compute the

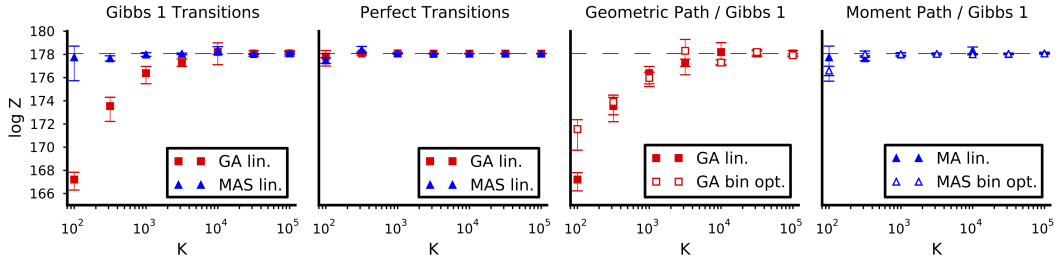

Figure 3: Estimates of $\log \mathcal{Z}_b$ for the tractable PCD(20) RBM as $K$, the number of intermediate distributions, is varied. Error bars indicate bootstrap 95% confidence intervals. (Best viewed in color.)

| $p_a(\mathbf{v})$ | path & schedule | CD1(20) | | | PCD(20) | | |
|---|---|---|---|---|---|---|---|
| | | $\log \mathcal{Z}_b$ | $\log \hat{\mathcal{Z}}_b$ | ESS | $\log \mathcal{Z}_b$ | $\log \hat{\mathcal{Z}}_b$ | ESS |
| uniform | GA linear | 279.59 | 279.60 | 248 | 178.06 | 177.99 | 204 |
| uniform | GA optimal binned | | 279.51 | 124 | | 177.92 | 142 |
| uniform | MAS linear | | 279.59 | **2686** | | 178.09 | **289** |
| uniform | MAS optimal binned | | 279.60 | **2619** | | 178.08 | **934** |

Table 1: Comparing estimates of $\log \mathcal{Z}_b$ and effective sample size (ESS) for tractable RBMs. Results are shown for $K = 100,000$ intermediate distributions, with 5,000 chains and Gibbs transitions. Bolded values indicate ESS estimates that are not significantly different from the largest value (bootstrap hypothesis test with 1,000 samples at $\alpha = 0.05$ significance level). The maximum possible ESS is 5,000.

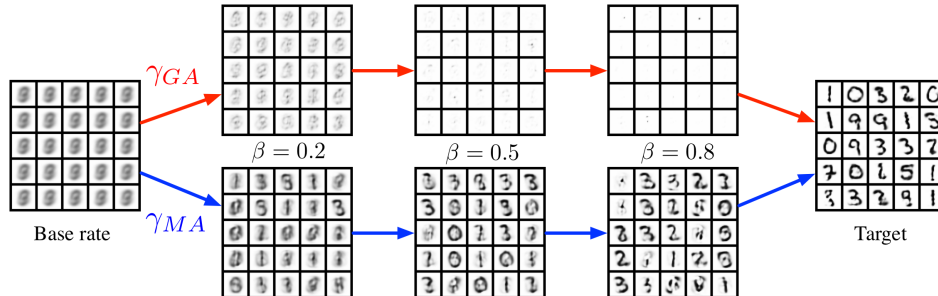

Figure 4: Visible activations for samples from the PCD(500) RBM. **(left)** base rate RBM, $\beta = 0$ **(top)** geometric path **(bottom)** MAS path **(right)** target RBM, $\beta = 1$.

partition function and moments and to generate exact samples by exhaustively summing over all $2^{20}$ hidden configurations. The moments of the target RBMs were computed exactly, and moment matching was performed with conjugate gradient using the exact gradients.

The results are shown in Figure 3 and Table 1. Under perfect transitions, $\gamma_{GA}$ and $\gamma_{MAS}$ were both able to accurately estimate $\log \mathcal{Z}_b$ using as few as 100 intermediate distributions. However, using the Gibbs transition operator, $\gamma_{MAS}$ gave accurate estimates using fewer intermediate distributions and achieved a higher ESS at $K = 100,000$. To check that the improved performance didn't rely on accurate moments of $p_b$, we repeated the experiment with highly biased moments.[2] The differences in $\log \hat{\mathcal{Z}}_b$ and ESS compared to the exact moments condition were not statistically significant.

**Full-size, Intractable RBMs:** For intractable RBMs, moment averaging required approximately solving two intractable problems: moment estimation for the target RBM, and moment matching. We estimated the moments from 1,000 independent Gibbs chains, using 10,000 Gibbs steps with 1,000 steps of burn-in. The moment averaged RBMs were trained using PCD. (We used 50,000 updates with a fixed learning rate of 0.01 and no momentum.) In addition, we ran a cheap, inaccurate moment matching scheme (denoted MAS cheap) where visible moments were estimated from the empirical MNIST base rate and the hidden moments from the conditional distributions of the hidden units given the MNIST digits. Intermediate RBMs were fit using 1,000 PCD updates and 100 particles, for a total computational cost far smaller than that of AIS itself. Results of both methods are

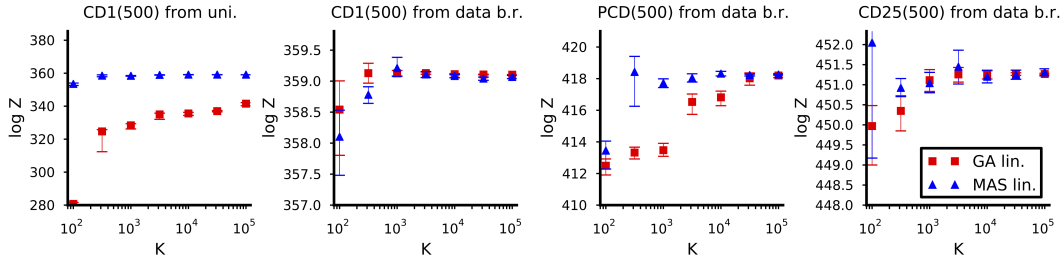

Figure 5: Estimates of $\log \mathcal{Z}_b$ for intractable RBMs. Error bars indicate bootstrap 95% confidence intervals. (Best viewed in color.)

| $p_a(\mathbf{v})$ | path | CD1(500) | | PCD(500) | | CD25(500) | |
| --- | --- | --- | --- | --- | --- | --- | --- |
| | | $\log \hat{\mathcal{Z}}_b$ | ESS | $\log \hat{\mathcal{Z}}_b$ | ESS | $\log \hat{\mathcal{Z}}_b$ | ESS |
| uniform | GA linear | 341.53 | 4 | 417.91 | 169 | 451.34 | 13 |
| uniform | MAS linear | 359.09 | 3076 | 418.27 | **620** | 449.22 | **12** |
| uniform | MAS cheap linear | 359.09 | **3773** | 418.33 | 5 | 450.90 | **30** |
| base rate | GA linear | 359.10 | **4924** | 418.20 | 159 | 451.27 | **2888** |
| base rate | MAS linear | 359.07 | 2203 | 418.26 | **1460** | 451.31 | 304 |
| base rate | MAS cheap linear | 359.09 | 2465 | 418.25 | 359 | 451.14 | 244 |

Table 2: Comparing estimates of $\log \mathcal{Z}_b$ and effective sample size (ESS) for intractable RBMs. Results are shown for $K = 100{,}000$ intermediate distributions, with 5,000 chains and Gibbs transitions. Bolded values indicate ESS estimates that are not significantly different from the largest value (bootstrap hypothesis test with 1,000 samples at $\alpha = 0.05$ significance level). The maximum possible ESS is 5,000.

shown in Figure 5 and Table 2. Overall, the MAS results compare favorably with those of GA on both of our metrics. Performance was comparable under MAS cheap, suggesting that $\gamma_{MAS}$ can be approximated cheaply and effectively. As with the tractable RBMs, we found that optimal binned schedules made little difference in performance, so we focus here on linear schedules.

The most serious failure was $\gamma_{GA}$ for CD1(500) with uniform initialization, which underestimated our best estimates of the log partition function (and hence overestimated held-out likelihood) by nearly 20 nats. The geometric path from uniform to PCD(500) and the moments path from uniform to CD1(500) also resulted in underestimates, though less drastic. The rest of the paths agreed closely with each other on their partition function estimates, although some methods achieved substantially higher ESS values on some RBMs. One conclusion is that it's worth exploring multiple initializations and paths for a given RBM in order to ensure accurate results.

Figure 4 compares samples along $\gamma_{GA}$ and $\gamma_{MAS}$ for the PCD(500) RBM using the base rate initialization. For a wide range of $\beta$ values, the $\gamma_{GA}$ RBMs assigned most of their probability mass to blank images. As discussed in Section 4.1, $\gamma_{GA}$ prefers configurations which are probable under both the initial and target distributions. In this case, the hidden activations were closer to uniform conditioned on a blank image than on a digit, so $\gamma_{GA}$ preferred blank images. By contrast, $\gamma_{MAS}$ yielded diverse, blurry digits which gradually coalesced into crisper ones.

## 6 Conclusion

We presented a theoretical analysis of the performance of AIS paths and proposed a novel path for exponential families based on averaging moments. We gave a variational interpretation of this path and derived an asymptotically optimal piecewise linear schedule. Moment averages performed well empirically at estimating partition functions of RBMs. We hope moment averaging can also improve other path-based sampling algorithms which typically use geometric averages, such as path sampling [12], parallel tempering [23], and tempered transitions [15].

## Acknowledgments

This research was supported by NSERC and Quanta Computer. We thank Geoffrey Hinton for helpful discussions. We also thank the anonymous reviewers for thorough and helpful feedback.

## Footnotes

[1]The ESS is defined as $M/(1 + s^2(w_*^{(i)}))$ where $s^2(w_*^{(i)})$ is the sample variance of the normalized weights [4]. In general, one should regard ESS estimates cautiously, as they can give misleading results in cases where an algorithm completely misses an important mode of the distribution. However, as we report the ESS in cases where the estimated partition functions are close to the true value (when known) or agree closely with each other, we believe the statistic is meaningful in our comparisons.

[2]In particular, we computed the biased moments from the conditional distributions of the hidden units given the MNIST training examples, where each example of digit class $i$ was counted $i + 1$ times.

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
