[Supplementary Material · supplementary.pdf]

# 1 Proof of Theorem 1

**Theorem 1.** *Suppose $K + 1$ distributions $p_k$ are linearly spaced along a path $\gamma$. Assuming perfect transitions, if $\boldsymbol{\theta}(\beta)$ and the Fisher information matrix $\mathbf{G}_{\boldsymbol{\theta}}(\beta) = \mathrm{cov}_{\mathbf{x} \sim p_{\boldsymbol{\theta}}}(\nabla_{\boldsymbol{\theta}} \log p_{\boldsymbol{\theta}}(\mathbf{x}))$ are continuous and piecewise smooth, then as $K \to \infty$ the bias $\delta$ behaves as follows:*

$$K\delta = K \sum_{k=0}^{K-1} \mathrm{D}_{\mathrm{KL}}(p_k \| p_{k+1}) \to \mathcal{F}(\gamma) \equiv \frac{1}{2} \int_0^1 \dot{\boldsymbol{\theta}}(\beta)^T \mathbf{G}_{\boldsymbol{\theta}}(\beta) \dot{\boldsymbol{\theta}}(\beta) \, \mathrm{d}\beta, \tag{1}$$

*where $\dot{\boldsymbol{\theta}}(\beta)$ represents the derivative of $\boldsymbol{\theta}$ with respect to $\beta$.*

*Proof.* First, assume that $\boldsymbol{\theta}(\beta)$ and $\mathbf{G}_{\boldsymbol{\theta}}(\beta)$ are both smooth. Consider a second-order Taylor expansion of $\mathrm{D}_{\mathrm{KL}}(\boldsymbol{\theta}(\beta) \| \boldsymbol{\theta}(\beta + h))$ around $h = 0$. The constant and first order terms are zero. For the second order term,

$$\nabla_{\boldsymbol{\theta}}^2 \mathrm{D}_{\mathrm{KL}}(\boldsymbol{\theta} \| \boldsymbol{\theta}_0)\big|_{\boldsymbol{\theta}=\boldsymbol{\theta}_0} = \mathbf{G}_{\boldsymbol{\theta}},$$

so the second-order Taylor expansion is given by:

$$\mathrm{D}_{\mathrm{KL}}(\boldsymbol{\theta}(\beta) \| \boldsymbol{\theta}(\beta + h)) = \frac{1}{2} h^2 \dot{\boldsymbol{\theta}}^T(\beta) \mathbf{G}_{\boldsymbol{\theta}}(\beta) \dot{\boldsymbol{\theta}}(\beta) + \epsilon,$$

where

$$|\epsilon| \leq \frac{h^3}{6} \max_{\beta} \left| \frac{d^3}{dh^3} \mathrm{D}_{\mathrm{KL}}(\boldsymbol{\theta}(\beta) \| \boldsymbol{\theta}(\beta + h)) \right|.$$

(The maximum is finite because $\mathbf{G}_{\boldsymbol{\theta}}$ is smooth.)

Assuming a linear schedule, the bias is given by

$$\begin{aligned}
\delta &= \sum_{k=0}^{K-1} \mathrm{D}_{\mathrm{KL}}(p_k \| p_{k+1}) \\
&= \sum_{k=0}^{K-1} \mathrm{D}_{\mathrm{KL}}(\boldsymbol{\theta}(k/K) \| \boldsymbol{\theta}((k+1)/K)) \\
&= \frac{1}{2K^2} \sum_{k=0}^{K-1} \dot{\boldsymbol{\theta}}(\beta_k)^T \mathbf{G}_{\boldsymbol{\theta}}(\beta_k) \dot{\boldsymbol{\theta}}(\beta_k) + \sum_{k=0}^{K-1} \epsilon_k
\end{aligned}$$

The error term decays like $1/K^2$, so it approaches zero even when scaled by $K$. The asymptotic bias, therefore, is determined by the first term. When scaled by $K$, this approaches

$$\mathcal{F}(\gamma) \equiv \frac{1}{2} \int_0^1 \dot{\boldsymbol{\theta}}(\beta)^T \mathbf{G}_{\boldsymbol{\theta}}(\beta) \dot{\boldsymbol{\theta}}(\beta) \, \mathrm{d}\beta.$$

Therefore, $K\delta \to \mathcal{F}(\gamma)$.

In the above analysis, we assumed that $\boldsymbol{\theta}(\beta)$ and $\mathbf{G}_{\boldsymbol{\theta}}(\beta)$ were smooth. If they are merely piecewise smooth, the integral decomposes into sums over the smooth segments of $\gamma$. Similarly, the KL divergence terms corresponding to non-smooth points decay like $1/K^2$, so they approach zero when scaled by $K$. Ignoring these terms, the bias decomposes as a sum over the smooth segments of $\gamma$, so the theorem holds in the piecewise smooth case as well. $\qquad \square$

# 2 Derivation of variational interpretation from Section 4.1

## 2.1 Geometric averages

For simplicity of notation, assume the state space $\mathcal{X}$ is discrete. Consider solving for a distribution $q$ to minimize the weighted sum of KL divergences

$$(1 - \beta) \mathrm{D}_{\mathrm{KL}}(q \| p_a) + \beta \mathrm{D}_{\mathrm{KL}}(q \| p_b) \tag{2}$$

with the constraint that $\sum_{\mathbf{x}} q(\mathbf{x}) = 1$. The Lagrangian is given by:

$$\mathcal{L}(q) = \lambda \left( \sum_{\mathbf{x}} q(\mathbf{x}) - 1 \right) + (1 - \beta) \sum_{\mathbf{x}} q(\mathbf{x}) \left( \log q(\mathbf{x}) - \log p_a(\mathbf{x}) \right) +$$

$$+ \beta \sum_{\mathbf{x}} q(\mathbf{x}) \left( \log q(\mathbf{x}) - \log p_b(\mathbf{x}) \right)$$

$$= -\lambda + \sum_{\mathbf{x}} \lambda q(\mathbf{x}) + q(\mathbf{x}) \log q(\mathbf{x}) - q(\mathbf{x}) \left[ (1 - \beta) \log p_a(\mathbf{x}) - \beta \log p_b(\mathbf{x}) \right]$$

Differentiating with respect to $q(\mathbf{x})$,

$$\frac{\partial \mathcal{L}(q)}{\partial q(\mathbf{x})} = \lambda + 1 + \log q(\mathbf{x}) - (1 - \beta) \log p_a(\mathbf{x}) - \beta \log p_b(\mathbf{x}).$$

Setting this to zero gives:

$$q(\mathbf{x}) \propto p_a(\mathbf{x})^{1-\beta} p_b(\mathbf{x})^{\beta}.$$

This is the optimum over the probability simplex. If $p_a$ and $p_b$ belong to an exponential family $\mathcal{P}$, with natural parameters $\boldsymbol{\eta}_{p_a}$ and $\boldsymbol{\eta}_{p_b}$, the optimum is achieved within $\mathcal{P}$ using $\boldsymbol{\eta}_{\beta} = (1 - \beta)\boldsymbol{\eta}_{p_a} + \beta\boldsymbol{\eta}_{p_b}$.

## 2.2 Moment averages

Suppose we wish to find

$$p_{\beta}^{(MA)} = \arg\min_q (1 - \beta) \mathrm{D}_{\mathrm{KL}}(p_a \| q) + \beta \mathrm{D}_{\mathrm{KL}}(p_b \| q). \qquad (3)$$

We write the cost function in terms of the natural parameters $\boldsymbol{\eta}$:

$$J(\boldsymbol{\eta}) = (1 - \beta) \sum_{\mathbf{x}} p_a(\mathbf{x})(\log p_a(\mathbf{x}) - \log q(\mathbf{x})) + \beta \sum_{\mathbf{x}} p_b(\mathbf{x})(\log p_b(\mathbf{x}) - \log q(\mathbf{x}))$$

$$= \mathrm{const} - \sum_{\mathbf{x}} \left[ (1 - \beta) p_a(\mathbf{x}) + \beta p_b(\mathbf{x}) \right] \log q(\mathbf{x})$$

$$= \mathrm{const} + \log \mathcal{Z}(\boldsymbol{\eta}) - \sum_{\mathbf{x}} \left[ (1 - \beta) p_a(\mathbf{x}) + \beta p_b(\mathbf{x}) \right] \boldsymbol{\eta}^T \mathbf{g}(\mathbf{x})$$

The partial derivatives are given by:

$$\frac{\partial J}{\partial \eta_i} = \sum_{\mathbf{x}} q(\mathbf{x}) g_i(\mathbf{x}) - \sum_{\mathbf{x}} \left[ (1 - \beta) p_a(\mathbf{x}) + \beta p_b(\mathbf{x}) \right] g_i(\mathbf{x})$$

$$= \mathbb{E}_q[g_i(\mathbf{x})] - (1 - \beta) \mathbb{E}_{p_a}(g_i(\mathbf{x})) - \beta \mathbb{E}_{p_b}(g_i(\mathbf{x}))$$

Setting this to zero, we see that the optimal solution is given by averaging the moments of $p_a$ and $p_b$:

$$\mathbb{E}_q[g_i(\mathbf{x})] = (1 - \beta) \mathbb{E}_{p_a}(g_i(\mathbf{x})) + \beta \mathbb{E}_{p_b}(g_i(\mathbf{x}))$$

Intuitively, this can be thought of as a maximum likelihood estimate of $\boldsymbol{\eta}$ for a dataset with $(1 - \beta)$ fraction of the points drawn from $p_a$ and $\beta$ fraction drawn from $p_b$.

## 3 Analysis of Gaussian example in Section 4.2

Here we evaluate the cost functionals for the Gaussian example of Section 4.2 under $\gamma_{GA}$ and $\gamma_{MA}$ using both linear and optimal schedules. Recall that $p_a = \mathcal{N}(\mu_a, \sigma)$ and $p_b = \mathcal{N}(\mu_b, \sigma)$. The natural parameters of the Gaussian are the information form representation, with precision $\lambda = 1/\sigma^2$ and potential $h = \lambda\mu$. The sufficient statistics are the first and (rescaled) second moments given by $\mathbb{E}[x] = \mu$ and $-\frac{1}{2}\mathbb{E}[x^2] = -\frac{1}{2}s \equiv -\frac{1}{2}(\sigma^2 + \mu^2)$.

Throughout this section, we use the relationship $\mathbf{G}_{\boldsymbol{\eta}}\dot{\boldsymbol{\eta}} = \dot{\mathbf{s}}$ (Amari (2000), sec. 3.3), so that $\mathcal{F}(\gamma)$ can be rewritten as $\frac{1}{2} \int_0^1 \dot{\boldsymbol{\eta}}(\beta)^T \dot{\mathbf{s}}(\beta) \, \mathrm{d}\beta$.

To simplify calculations, let $\beta$ range from $-1/2$ to $1/2$ (rather than 0 to 1), and assume $\mu_a = -1/2$ and $\mu_b = 1/2$. The general case can be obtained by rescaling $\mu_a$, $\mu_b$, and $\sigma$.

## 3.1 Geometric averages

Geometric averages correspond to averaging the natural parameters:

$$\lambda(\beta) = 1/\sigma^2$$
$$h(\beta) = \beta/\sigma^2$$

Solving for the moments,

$$\mu(\beta) = \beta$$
$$s(\beta) = \sigma^2 + \beta^2.$$

The derivatives are given by:

$$\dot{\lambda}(\beta) = 0$$
$$\dot{h}(\beta) = 1/\sigma^2$$
$$\dot{\mu}(\beta) = 1$$
$$\dot{s}(\beta) = 2\beta$$

Ignoring the constant, the cost functional is given by:

$$\mathcal{F}(\gamma) = \frac{1}{2} \int_{-1/2}^{1/2} \dot{h}(\beta)\dot{\mu}(\beta) - \frac{1}{2}\dot{\lambda}(\beta)\dot{s}(\beta) \, d\beta$$
$$= \frac{1}{2} \int_{-1/2}^{1/2} \frac{1}{\sigma^2} \, d\beta$$
$$= \frac{1}{2\sigma^2}.$$

We can also compute the cost under the optimal schedule by computing the path length (see Section 3):

$$\ell(\gamma) = \int_{-1/2}^{1/2} \sqrt{\dot{h}(\beta)\dot{\mu}(\beta) - \frac{1}{2}\dot{\lambda}(\beta)\dot{s}(\beta)} \, d\beta$$
$$= \int_{-1/2}^{1/2} \sqrt{1/\sigma^2} \, d\beta$$
$$= \frac{1}{\sigma}.$$

Since the functional under the optimal schedule is given by $\ell^2/2$, these two answers agree with each other, i.e. the linear schedule is optimal.

We assumed for simplicity that $\mu_a = -1/2$ and $\mu_b = 1/2$. In general, we can rescale $\sigma$ and $\mu_b - \mu_a$ by the same amount without changing the functional. Therefore, $\mathcal{F}(\gamma_{GA})$ is given by:

$$\frac{(\mu_b - \mu_a)^2}{2\sigma^2} \equiv \frac{d^2}{2}.$$

## 3.2 Moment averaging

Now let's look at moment averaging. The parameterizations are given by:

$$\mu(\beta) = \beta$$
$$s(\beta) = \sigma^2 + \frac{1}{4}$$
$$\lambda(\beta) = \left(\sigma^2 + \frac{1}{4} - \beta^2\right)^{-1}$$
$$h(\beta) = \left(\sigma^2 + \frac{1}{4} - \beta^2\right)^{-1}\beta$$

with derivatives

$$\dot{\mu}(\beta) = 1$$
$$\dot{s}(\beta) = 0$$
$$\dot{\lambda}(\beta) = 2\left(\sigma^2 + \frac{1}{4} - \beta^2\right)^{-2}\beta$$
$$\dot{h}(\beta) = \lambda(\beta)\dot{\mu}(\beta) + \mu(\beta)\dot{\lambda}(\beta)$$
$$= \left(\sigma^2 + \frac{1}{4} - \beta^2\right)^{-1} + 2\left(\sigma^2 + \frac{1}{4} - \beta^2\right)^{-2}\beta^2$$

The cost functional is given by:

$$\mathcal{F}(\gamma_{MA}) = \frac{1}{2}\int_{-1/2}^{1/2} \dot{\mu}(\beta)\dot{h}(\beta) - \frac{1}{2}\dot{s}(\beta)\dot{\lambda}(\beta)$$
$$= \frac{1}{2}\int_{-1/2}^{1/2} \dot{h}(\beta)\,\mathrm{d}\beta$$
$$= \frac{1}{2}[h(1/2) - h(-1/2)]$$
$$= \frac{1}{2\sigma^2}.$$

This agrees exactly with $\mathcal{F}(\gamma_{GA})$, consistent with Theorem 2.

However, we can see by inspection that for small $\sigma$, most of the mass of this integral is concentrated near the endpoints, where the variance changes suddenly. This suggests that the optimal schedule would place more intermediate distributions near the endpoints.

We can bound the cost under the optimal schedule by bounding the path length $\ell(\gamma_{MA})$:

$$\ell(\gamma_{MA}) = \int_{-1/2}^{1/2} \sqrt{\dot{\mu}(\beta)\dot{h}(\beta) - \frac{1}{2}\dot{s}(\beta)\dot{\lambda}(\beta)}\,\mathrm{d}\beta$$
$$= \int_{-1/2}^{1/2} \sqrt{\dot{h}(\beta)}\,\mathrm{d}\beta$$
$$= \int_{-1/2}^{1/2} \sqrt{\lambda(\beta)\dot{\mu}(\beta) + \mu(\beta)\dot{\lambda}(\beta)}\,\mathrm{d}\beta$$
$$\leq \int_{-1/2}^{1/2} \sqrt{|\lambda(\beta)\dot{\mu}(\beta)|}\,\mathrm{d}\beta + \int_{-1/2}^{1/2} \sqrt{|\mu(\beta)\dot{\lambda}(\beta)|}\,\mathrm{d}\beta$$
$$= \int_{-1/2}^{1/2} \frac{1}{\sqrt{\sigma^2 + \frac{1}{4} - \beta^2}}\,\mathrm{d}\beta + \sqrt{2}\int_{-1/2}^{1/2} \frac{|\beta|}{\sigma^2 + \frac{1}{4} - \beta^2}\,\mathrm{d}\beta$$
$$= 2\sin^{-1}\left(\frac{1}{\sqrt{4\sigma^2 + 1}}\right) + \sqrt{2}\log\left(1 + \frac{1}{4\sigma^2}\right)$$
$$\leq \pi + \sqrt{2}\log\left(1 + \frac{1}{4\sigma^2}\right)$$

The path length has dropped from linear to logarithmic! Since $\mathcal{F}$ grows like $\ell^2$, the cost drops from quadratic to $\log$ squared.

This shows that even though Theorem 2 guarantees that both $\gamma_{GA}$ and $\gamma_{MA}$ have the same cost under a linear schedule, one path may do substantially better than the other if one is allowed to change the schedule.