[Reviews · NeurIPS 2013]

Submitted by Assigned_Reviewer_5

AIS has become an important tool to evaluate the performance of a learning algorithm for MRFs. In common practice, people use geometric averaged models as intermediate models. This paper raises an interesting question of finding a better alternative to the regular annealing path. It focuses on the bias of the expected log-weight, on which the variance of the AIS output is monotonically related, and compares the asymptotical behavior of different annealing paths.

Overall, I think this paper is well written. The explanation is clear, the example of the two paths between Gaussian distributions in Figure 1 and the visualization of intermediate RBMs in Figure 4 are very helpful. Also the analysis is sound (I didn’t check the proof in the supplementary).

My only concern is about how to obtain the moment averaged models. The authors use PCD with a relatively short training time to do the moment matching. How sensitive is the outcome of AIS wrt the accuracy of the moments of the target distribution as well as the parameters of those intermediate models? How is the total computational cost of the new method compared to method based on geometric average?

In the geometric averaging path, it is observed for RBMs that assigning more models at the low temperature end usually obtains much smaller variance than the linear schedule in practice. Is this phenomenon considered or compared in the experiments?
Summary: This paper proposes an alternative to the regular geometric average annealing schedule for AIS, and shows a smaller variance and higher effective sample size on the output weights. The paper is well written and the result is convincing.

Submitted by Assigned_Reviewer_7

Summary
=======
Annealed importance sampling (AIS) depends on a sequence of distributions between an initial and a target distribution. While most previous applications of AIS have used distributions along paths in parameter space based on geometric averages of the two distributions, the present article proposes using alternative paths based on averaging moments. The authors provide theoretical and empirical evidence for the advantages of their approach.


Quality
=======
The paper gives a rigorous analysis of the proposed moment averaging (MA) path. The theoretical claims appear to be sound. Except for typographical errors, I did not find any mistakes in the proofs provided.

The empirical results on intractable models used a linear annealing schedule for the geometric averaging (GA) path. Since previous work seems to have found other schedules to be more effective (Neal, 1998; Salakhutdinov & Murray, 2008), it would be interesting to also see how well GA does with a different schedule when applied to the larger RBMs – despite or especially because of figures 2 and 3 suggesting that it will not make a difference. With the schedules used in the mentioned work, the sampler will spend more time close to the target distribution, so that I suspect that using them will at least lead to more reasonable samples in Figure 4.


Clarity
=======
The paper is well written and well structured. I had no problems following the main text.

The supplementary material might benefit from making a few more calculations explicit (for example, the Fisher information matrix used in the integral at the top of page 4).


Originality
===========
This paper represents a highly original contribution. As pointed out by the authors, virtually all previous applications of AIS used geometric paths. This shows that the proposed alternative is not at all obvious and technically challenging.


Significance
============
AIS is a widely used technique in statistical machine learning. While the MA approach seems to work reasonably well for estimating partition functions of some models (such as RBMs), getting it to produce useful results with more complex models can be difficult. Better annealing strategies have the potential to improve this situation and to allow for better model comparisons. In addition, as also mentioned by the authors, many sampling strategies important for inference and learning can benefit from better annealing strategies.

Although I am not convinced that the proposed annealing strategy will find widespread adoption (mainly because of the added complexity of having to estimate intermediate distributions and because the results on intractable RBMs suggest that GA can still compete with or even outperform MA, as measured by the ESS), I think it has great potential to inspire further work in this direction.


Minor comments
==============
– Supp. 1: better use $\theta_0 = \theta$ instead of $\theta = \theta_0$
– Supp. 2: better use $p_a$ and $p_b$ for $p_0$ and $p_1$ throughout
– Supp. 2.1: the first term of the Lagrangian should be $\lambda ( 1 - \sum_x q(x) )$ or $\lambda ( \sum_x q(x) - 1 )$
– Supp. 3.1: $\dot s(\beta) = t$ should be $\dot s(\beta) = 2\beta$
– Supp. 3.2: a factor 2 is missing in the derivative of $\lambda(\beta)$
Summary: This is a highly original contribution with potential future impact on learning, inference, and model comparison. While I am not convinced that it will find immediate adoption for the purpose of partition function estimation, I think it will inspire more theoretical work in the direction taken by the authors.

Submitted by Assigned_Reviewer_8

This paper presents an alternative method for choosing intermediate distributions when performing AIS on distributions in the exponential family. Experimental results are promising. These results may be of significant practical benefit, as they could improve researchers' ability to make objective comparisons between complex probabilistic models. Overall I liked it. I did think the theoretical motivation for the proposed intermediate distributions was somewhat lacking.

Equation 3 -- cool!

"high curvature" --> not high curvature. Rather, large information distance per theta distance.

Section 5.1 -- why is the larger variance of 1437.89 *better*? I would expect larger variance to be worse.

The experimental results were entirely for second order models, while the technique is proposed for arbitrary exponential family models. It would be nice to see experiments for other choices of sufficient statistic.

page 7 - "substantially", say something quantitative.

Some speculation follows:

It's never really explained why interpolating between the sufficient statistics should be better than interpolating between the parameter values.

Eqs. 4 and 5 strongly suggest that the optimal path in parameter space is the geodesic (minimal distance) path between the two parameterizations, with the Fisher information as a metric.

If I've done my algebra correctly, then $ds/d\theta = G$. (write down s = \int dx p(x) dE/dtheta, and take the gradient) If the path is linear in $s$, as you propose, then for small steps $\Delta s$, $\Delta \theta = d\theta / ds \Delta s = G^{-1} \Delta s$. An infinitesimal step along your proposed trajectory in terms of \theta looks like the step in s times the inverse Fisher information matrix. This is the same as moving towards the final \theta taking into account the information geometry (with metric = Fisher information matrix). That is, these updates have a functional form which is nearly identical to the natural gradient update.

Interpreting your algorithm in terms of a natural gradient trajectory also opens up some possible approximations/extensions. eg, roughly in order of decreasing promise:
- Apply the same technique to non-exponential family models, where the sufficient statistics are not defined, but the Fisher information still is.
- Use well motivated metrics other than the Fisher information -- the matrix natural gradient is likely to best bet here (Amari, S.-I. (1998). Natural Gradient Works Efficiently in Learning. Neural Computation)
- Use an approximation to G^-1, which can be computed much more cheaply (Sohl-Dickstein, J. (2012). The Natural Gradient by Analogy to Signal Whitening, and Recipes and Tricks for its Use. arXiv:1205.1828v1.)
- Iterate over choices for the entire trajectory, in order to find the global shortest trajectory (with metric G), rather than just taking the local shortest steps

--

Finally, here's a plug for reproducible science. You can become more awesome by putting source code for experiments in the supplemental material! When source code is shared for computational experiments, it has a major positive impact on the usefulness of the authors' hard work for future researchers, on the credibility of the work, and on the number of citations earned. And as a reviewer it makes me more positively disposed towards a paper.
Summary: These results may be of significant practical benefit, as they could improve researchers' ability to make objective comparisons between complex probabilistic models. Experimental results are promising, theoretical motivation is not fully satisfying.
Author Feedback

Author rebuttal: We thank all three reviewers for their enthusiastic and detailed feedback. We especially thank R8 for some promising suggestions of ways to build upon this work. We will release a software package so that the results can be reproduced.

R5 and R7 both ask whether geometric averages (GA) would perform better using alternative schedules. We believe the gains from this would be limited, for several reasons. First, we tried using the geometric schedule of (Neal, 1998), which is optimal in the specific case of scale parameters, but found that it worked consistently worse than the other schedules. Second, in our toy Gaussian example, moment averages (MA) outperform GA even when the optimal schedule is used. Third, the results in Figures 2 and 3 indicate that using the optimal binned schedules made little difference, and our findings were similar in all experiments we ran.

We note that our work was initially motivated towards eliminating the laborious hand-tuning of schedules that's required to get AIS to perform well, but as our work progressed, we found that the schedules made less difference than the path itself.

R5 and R7 ask about the cost and sensitivity of obtaining moments and fitting MA distributions. In the experiments with small RBMs (lines 371-372), we found that drastically perturbing the model moments made little difference to the results. Similarly, the experiments for intractable RBMs were repeated with small numbers of particles and PCD steps, and the quantitative results did not change much (lines 405-410). These results indicate that accurate MA models are not necessary for good performance.

R8 wants a clearer statement of the advantages of the MA path. We discussed three advantages in the paper: (1) the variational analysis of Section 4.1, which suggests that the MA distributions want to cover all relevant modes of the target distribution, in contrast with GA, which can miss important modes entirely. For algorithms based on importance sampling, it is essential that the proposal distribution cover all of the target distribution. (2) the optimal binned schedules of Section 4.3, which are effectively free because they can be easily computed from the computations already performed for MA; and (3) the Gaussian example of Section 4.2, where MA performs within a constant factor of the optimal path, suggesting that MA could result in large improvements over GA for some models. We will clarify these points in the final version.

R8 asks about the variance of the log weights of MA in Section 5.1. Because the MA intermediate distributions are more spread out (according to the variational analysis of section 4.1), some of the particles accurately model the target distribution, while others die out. In contrast, the GA particles are roughly equally bad. This is reflected in the larger variance of the log weights for MA compared to GA.

Finally, we very much appreciate R8's suggestions of new algorithms to investigate. R8 is correct that the optimal path according to (4) and (5) is the geodesic on the Riemannian manifold. Gelman and Meng (1998) used this to derive the optimal path for Gaussians (for path sampling, rather than AIS) as discussed in Section 4.2. Unfortunately, we don't know of an efficient way to compute or approximate the geodesic in general.

We also considered iterative approaches similar to what R8 suggests. Two issues would have to be overcome: first, even approximating G requires computing gradients (i.e. moments), which is often intractable. Second, the intermediate distributions in the path are coupled in the optimization, so convergence could be an issue. Our MA approach has the convenient features that the moments only need to be estimated once, and that each of the MA distributions can be fit independently since they only depend on the initial and target distributions and the optimization problems are convex. However, it's likely that studying these iterative approaches could also be fruitful.